# Intestinal Enteroendocrine Cells: Present and Future Druggable Targets

**DOI:** 10.3390/ijms24108836

**Published:** 2023-05-16

**Authors:** Roger Atanga, Varsha Singh, Julie G. In

**Affiliations:** 1Department of Internal Medicine, Division of Gastroenterology and Hepatology, University of New Mexico, Albuquerque, NM 87131, USA; ratanga@salud.unm.edu; 2Department of Medicine, Division of Gastroenterology and Hepatology, Johns Hopkins University School of Medicine, Baltimore, MD 21205, USA

**Keywords:** enteroendocrine cells, inflammatory bowel disease, metabolic disease, intestinal hormones, therapeutics

## Abstract

Enteroendocrine cells are specialized secretory lineage cells in the small and large intestines that secrete hormones and peptides in response to luminal contents. The various hormones and peptides can act upon neighboring cells and as part of the endocrine system, circulate systemically via immune cells and the enteric nervous system. Locally, enteroendocrine cells have a major role in gastrointestinal motility, nutrient sensing, and glucose metabolism. Targeting the intestinal enteroendocrine cells or mimicking hormone secretion has been an important field of study in obesity and other metabolic diseases. Studies on the importance of these cells in inflammatory and auto-immune diseases have only recently been reported. The rapid global increase in metabolic and inflammatory diseases suggests that increased understanding and novel therapies are needed. This review will focus on the association between enteroendocrine changes and metabolic and inflammatory disease progression and conclude with the future of enteroendocrine cells as potential druggable targets.

## 1. Introduction

The intestinal epithelia have a unique role in human health, as the interface between the luminal contents of the ‘outside’ and the systemic responses from the ‘inside’. Various intestinal epithelial cells (IECs) work in concert to maintain communication and proper function between the lumen and non-epithelial cells beneath the basolateral domain. Intestinal stem cells at the base of the crypts and the rapidly cycling transit amplifying cells are responsible for the rapid renewal of the intestinal epithelium and give rise to sub-populations of differentiated cells within the epithelial lining [1]. These cell types include enterocytes, the most prominent cell type of the intestinal epithelium that is responsible for nutrient and water absorption [2], various secretory cells such as goblet cells that secrete mucins [3], enteroendocrine cells that secrete hormones [4], and Paneth cells that release antimicrobial factors to protect nearby stem cells at the base of small intestinal crypts [5]. There are also chemosensory tuft cells [6] that play a key role in defense against helminths, and M cells that function in the uptake and transport of antigens from the lumen to specialized antigen-presenting cells in Peyer’s patches and lymphoid follicles [7].

With recent scientific advances, our appreciation of the diverse functions of IECs has dramatically expanded. Prime examples of this complexity are intestinal enteroendocrine cells (EECs) that comprise at least eight cellular subtypes. These subtypes are classified based on the hormones and/or peptides they produce and their localization along the crypt–villus axis [8]. Enteroendocrine cells, the focus of this review, are derived from the secretory lineage and distributed throughout the length of the small bowel and colon. These key sensory cells comprise approximately 1% of the epithelium but collectively form the largest endocrine system in humans [9]. Although many EEC subtypes exist [10], they are most commonly characterized as chromogranin A (CHGA)-positive cells [11]. These specialized cells function as sensors of luminal contents that then responsively secrete hormones and peptides from the basolateral surface. Short-chain fatty acids, other microbial metabolites, small peptides, amino acids, glucose, and fructose bind to specific apical receptors and can be transported through the EECs [12]. The secreted hormones or peptides can recruit immune cells and/or directly interact with spinal and vagal afferent neurons of the enteric nervous system [13]. Due to the systemic influence of these specialized cells and their secreted products, they have become an attractive therapeutic target against various metabolic, inflammatory, and pathophysiological diseases. This review will focus on current research investigating enteroendocrine cells as present and future druggable targets.

## 2. Classification of Enteroendocrine Cells

Based on their individual structure and position in the gastrointestinal epithelia, EECs are divided into two groups: “open type”, with a bottleneck shape and apical prolongation with microvilli facing the lumen, or “closed type”, located close to the basal membrane, not reaching the lumen of the gut and lacking microvilli [14,15]. Open EECs detect luminal contents directly through microvilli interaction with luminal contents whereas the closed EECs are activated indirectly either through neural or humoral pathways. Closed EECs can form basolateral projections called neuropods; they directly synapse with nerves of the enteric nervous system [13]. Both types accumulate their secretory products in cytoplasmic granules and release them by exocytosis at the basolateral membrane upon mechanical, chemical, or neural stimulation [16]. Although the molecular mechanisms that control secretion have not been fully characterized, there is a reliance on intracellular calcium [17,18] and cyclic AMP [19]. In the EEC subtype, enterochromaffin (EC), serotonin (5-HT) secretion is controlled by increased expression of the intracellular calcium receptor, ryanodine receptor 3 [18]. It is not yet known if this signaling pathway is ubiquitous to all EEC subtypes for hormone and peptide secretion or specific to enterochromaffin cells. 

There are at least eight described EEC subtypes; however, our current understanding of epithelial plasticity suggests that the subtypes have the ability to switch and secrete different hormones and peptides in response to luminal changes [20]. The current classification of EEC subtypes, their function, and their location in the digestive tract are listed in Table 1. 

## 3. Enteroendocrine Cells and Inflammatory Bowel Disease (IBD)

In addition to nutrient absorption in the gut, the epithelia act as a barrier to commensal and pathogenic bacteria. There is a delicate balance in the crosstalk between the gut microbiota and the immune cells that reside in the lamina propria below the barrier [46]. Barrier dysregulation disrupts the balance between tolerance and immunity and can lead to chronic dysfunctions, such as inflammatory bowel disease [47]. IBD has been closely associated with altered gut microbiota resulting from an excessive inflammatory response to commensals or to pathogenic microbes. EECs express chemosensory receptors such as Toll-like receptors (TLR1, TLR2, and TLR4) that respond to intestinal bacterial TLR ligands [48,49] and G-protein-coupled receptors (GPR40, GPR41, GPR43, GPR119, and GPR120) that respond to gut-microbiota-derived short-chain fatty acids (SCFAs) or long-chain fatty acids (LCFAs) [50,51]. In response to the binding of microbial metabolites on these apical receptors, EECs responsively produce specific peptides, hormones, and cytokines that can act on immune cells [52]. This suggests that EECs have an important, yet not fully understood role in gut inflammation. EECs may be able to relay intestinal dysbiosis into physiological adaptation [4,53,54,55]. Known changes to EEC subtypes and the changes in secreted hormone status in inflammatory diseases are summarized in Table 2.

### 3.1. Enteroendocrine Cells in Ulcerative Colitis (UC) and Crohn’s Disease (CD)

Inflammation in UC is localized to the colon but can occur throughout the small and large intestines in CD. Enteroendocrine hormones play a crucial role in the pathophysiology of inflammatory bowel diseases, both UC and CD; they can modulate intestinal epithelial barrier function in a transcellular and paracellular manner [63]. There is evidence that EECs produce pro-inflammatory interleukin (IL)-17C during CD and UC with a potential role in disease progression [64]. The EEC subtypes, L cells and EC cells increase in numbers in lymphocytic colitis [65], while secretion of peptide tyrosine tyrosine (peptide YY) from L cells has been shown to decrease in both UC and CD [60,66]. By contrast, peptide YY and glucagon-like peptide 1 (GLP-1) secretion, both from L cells, increase in terminal ileal CD [61,67]. The contrasting and oftentimes contradictory natures of these EEC hormones and peptides are likely due to their broad range of functions—ranging from anti-inflammatory and regenerative to the promotion of dysbiosis. Peptide YY and GLP-1 are produced and secreted from the same EEC subtype yet can sometimes have contrasting secretion patterns in CD. This may indicate variables not yet studied in the context of EECs, such as the severity of inflammation and stage of disease progression.

EEC peptides have shown evidence of anti-inflammatory properties, with studies in animal models of IBD indicating that glucagon-like peptide 2 (GLP-2), from L cells, can have a protective role in disease progression [68]. Likewise, an anti-inflammatory role of GLP-1 and GLP-1 agonists has been reported in humans and mice [69,70]. Clinical trials are underway for the use of GLP-2 analogs (teduglutide and apraglutide) for the treatment of CD and short bowel syndrome [71,72], with promising results thus far. Teduglutide has been approved and is currently in use. It is a recombinant GLP-2 analog with an amino acid mutation that allows it to have a longer half-life than endogenous GLP-2 [73] by preventing cleavage from dipeptidyl peptidase IV (DPPIV). By contrast, apraglutide contains four amino acid mutations, which allows it to be more resistant to cleavage by DPPIV, and thus has a longer half-life than teduglutide [74]. 

Further studies are required to provide direct evidence of the crosstalk between EECs and the immune system in disease progression and severity. The essential and varied roles EECs play in IBD pathogenesis are becoming increasingly important. Overall, EECs are interesting therapeutic targets for IBD treatment and a potential alternative to current immunosuppressive therapies, possibly utilizing currently available drugs for treating metabolic syndromes [63].

### 3.2. Enteroendocrine Cells and Celiac Disease 

Celiac disease is a small intestinal autoimmune disease triggered in a subset of genetically predisposed individuals by consumption of the indigestible portion of gluten, or gliadins [75]. Celiac disease manifests as a malabsorptive enteropathy and is associated with significant morbidity secondary to malabsorption. With the current lack of therapeutics, celiac diagnosis is a significant burden on the individual. Disease progression has been attributed to the activation of CD4+ T helper cells [76], but a recent study into this disease using intestinal epithelial organoids has found that the intestinal epithelial layer is an underestimated hotspot of celiac pathology [77].

Alterations in EECs and respective incretins have been reported in celiac disease [78,79]. The peptide hormone levels of cholecystokinin (CCK, from I cells), glucose-dependent insulinotropic polypeptide (GIP, from K cells), and GLP-1 (from L cells) are reduced in the blood of celiac disease patients. These peptides are thought to be the basis of pancreatic dysfunction that occurs during celiac disease progression [80]. By contrast, ghrelin from X/A-like cells [56] and other chromogranin A (CHGA)+ cells [59] increase in the duodenum of celiac disease patients and correlate with inflammation [81]. Additionally, plasma levels of 5-HT were significantly increased in celiac disease patients following consumption of a gluten-rich diet [82]. Increases in serum GLP-2 [57], somatostatin [58], plasma neurotensin [83], and oxyntomodulin [84] have also been reported in these patients. 

EC cells may play a direct role in the pathogenesis of refractory celiac disease, a rare and severe form of celiac disease that is characterized by the persistence of celiac disease symptoms during the initial (up to 12 months) phase of a gluten elimination diet [85]. An increase in EC cells is thought to promote inflammation by increasing interferon (IFN)-gamma production [59]. The number of EC cells doubles in the duodenum of celiac disease patients despite villus atrophy, and the expression of the rate-limiting enzyme for 5-HT synthesis, tryptophan hydroxylase (TPH1) was significantly higher as well [59]. 

## 4. Enteroendocrine Cells and Metabolic Diseases

Metabolic diseases encompass disorders in which the natural metabolism and metabolic processes are disrupted. Diabetes and obesity are the most common metabolic diseases that are not necessarily inherited. There is increasing evidence that enteroendocrine hormones have important roles in the pathogenesis of metabolic diseases, specifically diabetes and obesity. These diseases are rising globally and greatly disrupt the quality of life. A better understanding of metabolic diseases that may originate from the gut, as well as improved therapeutics and preventative measures are needed to control these diseases. Known changes to specific subtypes of EECs and their respective hormones in diabetes and obesity are summarized in Table 3.

### 4.1. Role of Enteroendocrine Cells in Diabetes

Diabetes is a common yet still complicated disease. Disease progression can be simplified as insufficient insulin production from the pancreas or insufficient cellular response to insulin [91]. Interestingly, bariatric surgery (Roux-en-Y gastric bypass, laparoscopic gastric banding) can be used to manage diabetes, control hyperglycemia, and reduce incidence of developing diabetes in obese individuals [92]. The intestinal endocrine system may have a large role in diabetes management and progression. 

Enteroendocrine K and L cells play a key role in regulating appetite and glucose homeostasis. GLP-1 potentiates postprandial insulin secretion, delays gastric emptying, and inhibits glucagon secretion [93]. In the small intestine of diabetic subjects, there was a notable decrease in the number of CHGA+ cells, but an increase in the expression of peptide YY and GCG, both secreted by L cells, in the colon [90]. The variable responses from EECs between the small and large intestine during metabolic disease states have yet to be fully understood. 

A recent study has emphasized the role of GLP-1 in the pathogenesis of type II diabetes (T2D) in obese patients [89]. Using RNA sequencing, they reported a decrease in GLP-1 cell lineage, GLP-1 maturation from proglucagon, and a significant reduction of GLP-1 positive cells in jejunal samples of obese subjects with T2D compared to obese individuals who do not have T2D. GLP-1 acts on the pancreas to control insulin secretion and GLP-1-based therapies are now FDA (Food and Drug Administration)-approved for the treatment of T2D [94,95], however, patients with multiple metabolic diseases are likely to have a complicated EEC response to the available therapeutics. 

### 4.2. Enteroendocrine Cells and Obesity

Obesity, defined as excessive fat accumulation and body mass index over 30, is a major factor in the development of metabolic syndrome. Lifestyle and dietary changes are important considerations, yet the rate of obesity in adults and children continues to rise. A better understanding of the molecular changes in the gut that arise from obesity is needed to develop better and more efficient therapeutics. 

EECs contain nutrient receptors such as free fatty acid receptors (FFARs), calcium-sensing receptors (CaSRs), and taste receptors (TR) on their luminal surface which bind nutrients and trigger hormone release [96,97,98]. These nutrient-responsive hormones act on cells and neuronal pathways to modulate food intake, digestion, energy balance, and body weight. The potential importance of EECs is notably demonstrated in insulin resistance, diabetes, and obesity, which have all been associated with altered levels of EEC hormones [90,99]. Ghrelin (X/A-like cells) stimulates food intake and hunger sensation and decreases energy expenditure; this ultimately promotes weight gain [99], and is indicative of a positive correlation between obesity and ghrelin secretion. By contrast, obesity that is related to insulin resistance inhibits ghrelin secretion. In this specific metabolic disease, X/A-like cells express insulin receptors and insulin can directly suppress ghrelin secretion [100]. 

Cholecystokinin (CCK) from I cells promotes satiety and reduces food intake by suppressing hunger [101]; however, the role of CCK in obesity remains controversial. CCK-producing cells sense free fatty acids using fatty acid transporters FFAR1, FFAR4, and CD36; calcium-sensing receptors CaSR; lysophosphatidic acid receptor 5, LPAR5; and taste receptors TAS1R1/TAS1R3 [102]. While some studies have shown a delayed CCK response in obese patients following consumption of high-fat diets [103], there is also evidence that CCK release is elevated in obese subjects compared to lean individuals [86] or not changed at all [104]. The discordance in these findings may be due to differences in meal composition, study duration, and microbiota. Further exploration of these differences will open up the possibility of CCK-targeted enteroendocrine therapeutics that can promote satiety and reduce the burden of obesity. 

GLP-1 has been shown to have a role in regulating glucose, food intake, and promoting weight loss [105]. GLP-1 stimulates insulin release following glucose absorption and inhibits glucagon release. Obesity has been associated with low fasting GLP-1 levels and decreased postprandial GLP-1 response under conditions of dysregulated glycemic control [87]. However, there is increased expression and secretion of GLP-1 in obese patients with normal glycemia. This may be in part due to a compensatory mechanism to prevent the development of T2D [106]. Several GLP-1 analogs and receptor agonists that are effective in the treatment of T2D are being tested for efficacy in treating obesity as well [107]. 

One of the functions of peptide YY includes the inhibition of glucose-stimulated insulin release. Peptide YY is primarily stimulated by lipids, rather than proteins or glucose [102]. It regulates weight through the suppression of caloric consumption and increased fat oxidation [108]. Lower baseline levels of peptide YY are associated with obesity and an overall higher body mass index [109]. Chronic administration of peptide YY in murine models of diet-induced obesity led to decreased body weight, food intake, and overall improved insulin sensitivity [88]. Currently, peptide YY analogs are being explored in clinical trials for the management of obesity and T2D treatment [110]. 

## 5. The Role of Enteroendocrine Cells in Gut Injury

Enteroendocrine cells can be dynamically altered during injury to the gut and other organs. There is evidence that implicates EECs as facultative stem cells that promote plasticity and regeneration [111,112], but they can also contribute to increases in small intestinal neuroendocrine tumors [113]. However, other studies have shown that active, mature EECs are needed for the release of endocrine hormones necessary in halting injury or promoting healing and regeneration. To date, the regenerative role of EECs has been best characterized by ghrelin secretion from X/A-like cells. In a murine model of radiation-induced injury, ghrelin treatment promoted epithelial healing and attenuated inflammation via activation of Notch signaling [114]. Ghrelin has also been found to attenuate intestinal ischemia-induced injury in mice by activating the mammalian target of rapamycin (mTOR) signaling [115], as well as decrease sepsis-induced lung injury by blocking AKT, inducible nitric oxide synthase (iNOS), and nuclear factor kappa B (NF-kB) signaling in alveolar macrophages [116]. Overall, increased ghrelin secretion has an anti-inflammatory effect on injured tissue. 

GLP-1 is secreted in response to the presence of lipopolysaccharides (LPS) following intestinal barrier injury and protects against inflammation in mice [49]. However, this functional mechanism of regeneration via GLP-1 is not understood. Similarly, peripheral 5-HT circulates systemically by platelets and acts as a mitogen in injured organs. In murine models of liver injury, 5-HT bound to 5-HT2A and 2B receptors and promotes liver regeneration after partial hepatectomy [117]. Mice lacking TPH1 needed for 5-HT synthesis were unable to fully regenerate their liver. Although the connection between platelet-derived 5-HT, liver regeneration, and intestinal EECs was not explored in this study, these findings suggest that EECs can undergo adaptive reprogramming during injury and provides insights for potential new regenerative therapeutics.

## 6. Enteroendocrine Cells in the Gut–Brain Axis

The secretory products from EECs can reach distant targets through release in the bloodstream or act directly on nerve endings close to the site of release. The first report of EECs having paracrine functions was by Larsson et al., which described basal cytoplasmic prolongation in somatostatin-containing D cells [118]. Similar cytoplasmic prolongation processes have been reported in L cells containing peptide YY in addition to D cells [13,119]. Recent advances have suggested EEC sensory transmission by showing direct connections between EECs and the nervous system through which EECs can directly communicate with the neurons innervating the gastrointestinal tract to initiate gut–brain communication [120,121]. In this process, EECs contribute to the brain–gut axis by exerting regulatory effects between the brain and the gut. Vagal afferent terminals innervate the wall of the gastrointestinal tract and are close to the mucosal epithelium [122]. In response to peptides or hormones secreted by EECs, receptors located along the vagal afferent fibers convey stimuli generated by EECs to the brainstem [123].

Recent advances have described a pathogenic role of the microbiota–gut–brain axis in neurological and psychiatric disorders including Parkinson’s disease (PD) and schizophrenia, in which EECs might participate [124,125]. PD is a debilitating neurodegenerative disease characterized by motor disturbances, including resting tremors, rigidity, and slow movements, as well as gastrointestinal symptoms, such as constipation and gastroparesis [126]. It was recently discovered that α-synuclein (a major component of the pathological hallmarks of PD) is expressed in the EECs of mouse and human intestines [127]. Furthermore, α-synuclein-containing EECs directly connect to α-synuclein-containing nerves, forming a neural circuit between the gut and the nervous system. It is possible that toxins or other environmental influences in the gut lumen could affect α-synuclein folding in EECs, thereby initiating a process by which misfolded α-synuclein could propagate from the gut to the brain. Similarly, EECs are predicted to be involved in another major neurological disorder, schizophrenia. Importantly, serotonergic dysfunction is thought to contribute to the pathophysiology of schizophrenia [128,129]. An elevated level of prefrontal 5-HT1A receptors and a reduction in prefrontal 5-HT2A receptors have been observed in schizophrenia in a meta-analysis of post-mortem studies [130]. Since the largest reserve of 5-HT is produced in enterochromaffin cells [131] (Table 1), studies are underway to determine if systemically circulating 5-HT can influence neuronal receptors.

The EEC–neuron connection has opened an exciting new perspective on the complexity of the bidirectional communication between the brain and the gut. Targeting EEC receptors on the luminal side can potentially be used to activate hormonal and neuronal pathways providing a novel approach to treating neurological diseases. Furthermore, drugs changing EEC functions may participate in the control of depression, anxiety, and visceral hypersensitivity. 5-HT, a primary signaling neurotransmitter in CNS, gets altered in psychotic disorders, including schizophrenia, thus constituting a potential target of second-generation antipsychotics. Evidence suggests that activation of 5-HT receptors, specifically 5-HT3 and 5-HT4, sends signals from the gut to the CNS, which activates multiple pathways including those involved in nociception [132]. Recently, a significant amount of attention has been given to EECs and their role in visceral pain perception [133,134]. Increased understanding of the mechanistic role that EECs play in visceral pain may lead to novel non-addictive therapeutics that target specific EEC receptors. 

## 7. Current Therapeutics Targeting Enteroendocrine Hormones and Peptides

Short bowel syndrome (SBS) is a chronic malabsorption disorder in which the body is unable to absorb enough nutrients from food or drinks, generally from surgical removal of portions of the intestine [135]. GLP-2 analogs that target the GLP-2 receptor are currently used in the clinic to successfully treat SBS. These analogs are recombinant GLP-2 with amino acid mutations to prevent cleavage by DPPIV. GLP-2 administration in SBS patients greatly increases intestinal absorption of nutrients and water, reduces gastric emptying, and increases body weight and lean body mass [136]. These trophic effects of GLP-2 are likely due to stimulation of enterocyte proliferation and decreased enterocyte apoptosis [136,137], although further mechanistic studies are needed. Recent studies have conclusively found that semaglutide (a GLP-1 receptor agonist) mimics the action of GLP-1 by stimulating insulin release after a meal, prolonging satiety, and reducing glucose production by the liver [95]. Due to its mechanism of action on insulin release, it has been approved for the treatment of T2D. Additionally, GLP-1 receptor agonists are showing promising results in Phase 2 and 3 trials for the treatment of obesity [138,139]. One formulation of semaglutide, Wegovy, has recently been FDA-approved for treating obesity in adults and children over 12 years of age [140]. Although GLP-1 receptor agonists are useful for treating obesity, how will long-term use of these formulations affect the metabolic status of the intestinal epithelia and immune cells? As noted in Table 2 and Table 3, GLP-1 and 2 are often found with abnormal, increased secretion patterns in a variety of metabolic and inflammatory diseases. Therefore, further mechanistic studies on the systemic and long-term effects of GLP-1 receptor agonists are still needed. A potentially less physiologically disruptive approach to obesity treatment is supplementation with medium-chain fatty acids (MCFA) or their agonists. A recent study used MCFA agonists to activate MCFA colonic receptors, GPR84 and FFAR4, in obese adults [141]. These subjects experienced increased satiety and had higher levels of peptide YY but did not have significantly altered levels of GLP-1. The complexity of these hormones and peptides, particularly those secreted from the same EEC subtype, needs further exploration in this second wave of therapeutics for metabolic and inflammatory diseases. 

## 8. Conclusions

Enteroendocrine cells are being recognized for their roles in various metabolic, inflammatory, and autoimmune diseases. However, the secreted hormones and peptides ofttimes have contradictory—beneficial and detrimental—roles in these diseases. More studies are needed to understand the exact mechanistic role of these specialized cells in driving disease states or influencing healing and regeneration. Future therapeutics may target the EECs in attempts to manipulate the microbiota or neurological pathways. The interaction between EECs, immune cells, and enteric nerves highlights their importance in the systemic spread of diseases that may originate in the gut. Thus, enteroendocrine cells and their hormones and peptides are promising therapeutic targets for diseases described in this review. 

## Figures and Tables

**Table 1 ijms-24-08836-t001:** Known classifications of enteroendocrine cells.

Subtype	Hormones or Peptides	Receptors	Location	Main Function	References
EC cells	Serotonin (5-HT)	FFARs 2,3; TRPA1; TLRs; toxin receptors	Small intestine and colon	Gut motility and secretion, nausea, and vomiting	[21,22,23]
L cells	GLP-1, GLP-2, peptide YY	T2Rs; T1R2-T1R3; FFARs 1–3; GPR119, LPAR5, GPR120; CaSR	Ileum, colon	Appetite control, insulin release, slows intestinal transit, and stimulates cell proliferation	[24,25,26]
X/A-like cells	Ghrelin, nesfatin-1	T1R1-T1R3; T2Rs	Stomach	Appetite control and growth hormone release	[27,28]
K cells	GIP, 5-HT	GPR119, GPR120; FFAR1	Proximal small intestine	Stimulate insulin release and regulates gut mucosal growth	[29,30]
I cells	Cholecystokinin, 5-HT	T2Rs; FFA1; GPR120; CCK1R; CCK2	Proximal small intestine	Stimulate pancreatic enzyme secretion and activate bile release	[31,32]
D cells	Somatostatin	GPCRs	Stomach, small intestine	Inhibit gastric acid secretion, motility, and intestinal absorption	[33,34]
G cells	Gastrin	CCK2R	Stomach, duodenum	Regulate gastric acid secretion and epithelial proliferation	[35,36,37]
N cells	Neurotensin	NTSR1, NTSR2	Small intestine	Stimulate pancreatic and biliary secretions and suppress small-bowel gastric motility	[38,39]
M cells	Motilin	PAFR, TLR-4, α5β1-integrin	Small intestine, colon, rectum	Regulate the migrating motor complex and increase insulin release	[40,41]
S cells	Secretin	SR, SecR	Stomach, small intestine	Regulate pancreatic exocrine secretion, gastric acid secretion, and gastric motility	[42,43]
ECL-like cells	Histamine	H1R, H2R, H3R	Stomach	Stimulate gastric acid (HCl) and pepsinogen secretion	[44,45]

**Table 2 ijms-24-08836-t002:** Summary of changes to enteroendocrine cells and secreted hormones or peptides in inflammatory diseases.

Pathophysiology	Specific EE Subtypes Changed	Hormone or Peptide Changed	Reference
Celiac disease	X/A-like cells	↑ Ghrelin	[56]
Celiac disease	L cells	↑ GLP-2	[57]
Celiac disease	D cells	↑ SST	[58]
Refractory Celiac disease (RCD)	EC cells	↑ TPH1	[59]
Crohn’s disease	L cells	↓ PYY	[60]
Terminal ileal Crohn’s disease	L cells	↑ GLP-1, PYY	[61]
Ulcerative colitis	L cells	↑ GLP-1	[62]
Diverticulitis	L cells	↓ GLP-1	[62]
Gut inflammation via LPS	L cells	↑ GLP-1	[49]

EE, enteroendocrine; EC, enterochromaffin; LPS, lipopolysaccharide; GLP, glucagon-like peptide; SST, somatostatin; TPH1, tryptophan hydroxylase 1; PYY, peptide tyrosine tyrosine. Red up arrows indicate an increase, blue down arrows indicate a decrease.

**Table 3 ijms-24-08836-t003:** Summary of changes to enteroendocrine cells and secreted hormones or peptides in metabolic diseases.

Pathophysiology	Specific EE Subtypes Changed	Hormone or Peptide Changed	Reference
Obesity	I cells	↑ CCK	[86]
Obesity	L cells	↓ GLP-1	[87]
Obesity	L cells	↓ PYY	[88]
Obesity-induced Type II diabetes	L cells	↓ GLP-1	[89]
Type II diabetes	K cells, L cells	↑ GIP, GCG, PYY	[90]

EE, enteroendocrine; CCK, cholecystokinin, GLP, glucagon-like peptide; PYY, peptide tyrosine tyrosine; GIP, gastric inhibitory peptide; GCG, proglucagon. Red up arrows indicate an increase, blue down arrows indicate a decrease.

## Data Availability

No new data were created or analyzed in this study. Data sharing is not applicable to this article.

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
