# Peer review of "Intestinal Enteroendocrine Cells: Present and Future Druggable Targets"

_ijms, 2023, doi:10.3390/ijms24108836_

Round 1
Reviewer 1 Report
This review deals with the importance of enteroendocrine cells as present and future druggable targets focusing on the association between enteroendocrine changes and the progression of some diseases. The subject is briefly described but, at the same time, updated and well documented. The manuscript is interesting and clearly explained.
However, some concepts could be further explored. In particular, I suggest that the authors discuss more deeply the factors that regulate the mechanisms underlying intestinal enteroendocrine cell secretion and expand the section on the role of enteroendocrine cells in the gut-brain axis (paragraph 6).
The following minor points are suggested:
- Please check the text carefully as there are some grammatical and/or typo errors
- Please check the References style following the instruction of the IJMS
- Abbreviations should be placed the first time the word appears and should be used throughout the text
- I would move table 3 after subparagraph 4.2 for a better reading of the text
Author Response
We thank reviewers for their efforts on this review. Accordingly, we have modified the review and are submitting the modified manuscript. Please find the point-by-point response to the reviewers’ concerns below:
I suggest that the authors discuss more deeply the factors that regulate the mechanisms underlying intestinal enteroendocrine cell secretion and expand the section on the role of enteroendocrine cells in the gut-brain axis (paragraph 6).
We have included the known secretion mechanisms based on available literature. Please note that the field of enteroendocrine cell is still evolving and detail mechanistic understanding of regulation of enteroendocrine cell secretion is still lacking.
The following minor points are suggested:
-Please check the text carefully as there are some grammatical and/or typo errors
- Please check the References style following the instruction of the IJMS
-Abbreviations should be placed the first time the word appears and should be used throughout the text
- I would move table 3 after subparagraph 4.2 for a better reading of the text
We thank the reviewer for careful reading of our manuscript and pointing out the grammatical and typo errors, including a number of improper abbreviations. We have carefully revised and proofread this resubmission to address and fix these concerns. The reference style has been updated to the IJMS style. Table placements will be amended when the manuscript is re-formatted in the IJMS template.
Reviewer 2 Report
I would like to begin by expressing my sincere appreciation for your efforts in attempting to address such a vast and complex subject matter. I understand the significant challenges that come with such an undertaking, and I commend you for taking on this task.
However, with all due respect, I must express my concern that the current state of your manuscript may not have fully addressed the subject matter to the extent required. In my humble opinion, the review produced may not be sufficient to facilitate the advancement of new research in this field.
As a humble suggestion, I kindly recommend that the focus of the manuscript be centered more closely on its intended objective. Additionally, it may be helpful to review the subject matter with more detail and attention, ultimately resulting in a manuscript that serves as a valuable resource for the collective research community.
Please know that my intentions are only to provide constructive feedback to support the betterment of your work. Thank you for your time and consideration, and I wish you all the best in your future endeavors.
Author Response
We thank reviewers for their efforts on this review. Accordingly, we have modified the review and are submitting the modified manuscript.
As a humble suggestion, I kindly recommend that the focus of the manuscript be centered more closely on its intended objective. Additionally, it may be helpful to review the subject matter with more detail and attention, ultimately resulting in a manuscript that serves as a valuable resource for the collective research community.
We respectfully disagree with this comment from the reviewer. We believe that we have created a focused review where we are presenting the available literature about therapeutic aspects of enteroendocrine cells in metabolic and inflammatory diseases that affect the intestine.
Reviewer 3 Report
This review was focus “on the association between enteroendocrine changes and metabolic and inflammatory disease progression and conclude with the future of enteroendocrine cells as potential druggable targets.”
The paper deals with an important topic, but I miss the emphasis on the originality of this study. The topic is not new and many review papers are already available. The authors should give their "findings" in the form of a Figure/Scheme at the end of the paper.
Many sentences are single and quotes taken from other review papers (citations from 1-7; 37-50). Sometimes two review papers (e.g., lines 44-45; 78-79) or even 3 papers (lines 88-89) are cited for a single sentence about "general" news. it is a bit like listing, rather than knowing the exact mechanism of action of a particular hormone produced by EECs. Please expand on this a bit.
Table 1 - citations to this data are missing (unless you can't see in the uploaded version), please complete the list of citations in the last column and add a legend under the table with an explanation of the abbreviations of the peptides/proteins in question.
Similarly, Table 2 - it seems that in the given diseases the main changes are related to GLP-1/2. Please comment on this in more detail in the text (what is the mechanism of this process) and also complete the legend under the table with abbreviations of peptides and colored arrows. Table 3 - please complete the legend with abbreviations, and align the content of the columns (peptides). I also recommend supplementing the paper, with a new Table (No. 4) under Chapter 7; with an inventory and mechanism of action of potential 'drugs' in the treatment of diseases in which there are gripping changes with the involvement of EECs, potential targets in the context of changes in hormone production. There are too many generalizations in the text (line 112-113; 177-178; 207-208; several GLP-1 analogues? what types?, line 216-218, ref. 70 and ref. 74 of PYY analogues).
The work is based on older publications, works from the last 5 years are less than 30%. And most of these recent papers, are also review papers. It is more than 55% of reviews papers cited.
Please improve the literature list according to the editorial board's requirements, especially with a large number of authors (ref. 23, 52, 55, 72, 102, etc.).
Minor errors I noticed:
in Table 1 - should be "celiac disease" and not "celiac" itself; also, everywhere in the text, please use the correct name of the disease;
line 100-101 - PYY - please correct the abbreviation explanation, it should be better "peptide YY"; line 145 - it should be "tryptophan"; Please clarify the abbreviation 'EE' in the tables and in the text.
line - 170 - abbreviation of GCG, please leave the abbreviation alone, as it is not clear from the cited paper that it is about preproglucagon (ref. 56). It could be about glucagon itself or GLP.
line 210 - use the word "glucose" rather than "blood sugar"
To summarize:
The paper needs additions, especially the addition of a table of potential therapies (in keeping with the title of the paper) in Chapter 7 and at least 1 figure with novelty of the review, and an emphasis on the novelty of this study (including in the conclusions).
Author Response
We thank reviewers for their efforts on this review. Accordingly, we have modified the review and are submitting the modified manuscript. Please find the point-by-point response to the reviewers’ concerns below:
The paper deals with an important topic, but I miss the emphasis on the originality of this study. The topic is not new and many review papers are already available. The authors should give their "findings" in the form of a Figure/Scheme at the end of the paper.
We respectfully partially disagree with this comment from the reviewer. We agree that there are many review papers available on enteroendocrine cells. The majority are focused on the classification of subtypes or a specific disease. We believe that we have created a focused review where we are presenting the available literature about current and potential therapeutic aspects of enteroendocrine cells.
Many sentences are single and quotes taken from other review papers (citations from 1-7; 37-50). Sometimes two review papers (e.g., lines 44-45; 78-79) or even 3 papers (lines 88-89) are cited for a single sentence about "general" news. it is a bit like listing, rather than knowing the exact mechanism of action of a particular hormone produced by EECs. Please expand on this a bit.
We have updated the references to what is most relevant and strived to convey the vast literature in our own words, rather than re-stating what others have written.
Table 1 - citations to this data are missing (unless you can't see in the uploaded version), please complete the list of citations in the last column and add a legend under the table with an explanation of the abbreviations of the peptides/proteins in question. Similarly, Table 2 - it seems that in the given diseases the main changes are related to GLP-1/2. Please comment on this in more detail in the text (what is the mechanism of this process) and also complete the legend under the table with abbreviations of peptides and colored arrows. Table 3 - please complete the legend with abbreviations, and align the content of the columns (peptides). I also recommend supplementing the paper, with a new Table (No. 4) under Chapter 7; with an inventory and mechanism of action of potential 'drugs' in the treatment of diseases in which there are gripping changes with the involvement of EECs, potential targets in the context of changes in hormone production. There are too many generalizations in the text (line 112-113; 177-178; 207-208; several GLP-1 analogues? what types?, line 216-218, ref. 70 and ref. 74 of PYY analogues).
We apologize for the not including the citations for Table 1. We have updated and included all relevant citations and defined abbreviations presented in the table. We comment in the text the discrepancies in GLP-1 expression and/or secretion that have been found in other studies. There is not yet enough data available to detail on the mechanisms related to this, however, we speculate that interactions with the microbiota or immune cells may be contributing factors. We have also expanded on the few known therapeutics that target enterohormones in the text.
The work is based on older publications, works from the last 5 years are less than 30%. And most of these recent papers, are also review papers. It is more than 55% of reviews papers cited.
We included relevant literature for this focused review. Much of the initial observations and discoveries related to enteroendocrine cells and enterohormones are 20-30 years old, however, they are relevant to our current and potential understanding of therapeutics targeting these cells. The more recent publications are focused on transcription factors and differentiation cues that are important knowledge, but out of the scope of this review article. We have updated our references to cite more primary research articles.
Please improve the literature list according to the editorial board's requirements, especially with a large number of authors (ref. 23, 52, 55, 72, 102, etc.).
We have formated the references according to IJMS style.
Minor errors I noticed:
in Table 1 - should be "celiac disease" and not "celiac" itself; also, everywhere in the text, please use the correct name of the disease; line 100-101 - PYY - please correct the abbreviation explanation, it should be better "peptide YY"; line 145 - it should be "tryptophan"; Please clarify the abbreviation 'EE' in the tables and in the text; line - 170 - abbreviation of GCG, please leave the abbreviation alone, as it is not clear from the cited paper that it is about preproglucagon (ref. 56). It could be about glucagon itself or GLP; line 210 - use the word "glucose" rather than "blood sugar".
We thank the reviewer for careful reading of this manuscript and these helpful critiques. We have proofread our revised manuscript to amend the grammatical and tyopgraphical errors and included the correct scientific verbiage suggested by the reviewer.
To summarize: The paper needs additions, especially the addition of a table of potential therapies (in keeping with the title of the paper) in Chapter 7 and at least 1 figure with novelty of the review, and an emphasis on the novelty of this study (including in the conclusions).
We have included more details in the text on the few therapies currently available. We have included a graphical abstract/figure that summarizes this review.
Round 2
Reviewer 1 Report
I have read the revised version of the manuscript and appreciate the effort made by the Authors to improve it.
I only recommend checking again the use of some abbreviations in the text (e.g. enteroendocrine cells are first mentioned in line 35; gastrointestinal (line 99) is only then abbreviated as GI (line 631).
Author Response
I only recommend checking again the use of some abbreviations in the text (e.g. enteroendocrine cells are first mentioned in line 35; gastrointestinal (line 99) is only then abbreviated as GI (line 631).
We thank the reviewer for their careful reading of our revised manuscript. We have checked abbreviations to ensure continuity.
Reviewer 3 Report
I have read the submitted manuscript after review and I believe that in its current form the work can be accepted for publication. It seems that all my comments have been taken into account and the work is now presented in a more professional manner.
The authors wrote in reply „We have included more details in the text on the few therapies currently available. We have included a graphical abstract/figure that summarizes this review”.
Unfortunately, I don't see the figure attached to the paper. Is it possible to show it?
Maybe it is contained in another file or sent separately? Either way, I think the work has been significantly revised, supplemented with more original work, and in this version it can be published.
If there were any minor proofreading errors (extra spaces, missing periods, etc.) that I did not notice, I hope they will "disappear" after the final brush-up.
Author Response
I have read the submitted manuscript after review and I believe that in its current form the work can be accepted for publication. It seems that all my comments have been taken into account and the work is now presented in a more professional manner.
The authors wrote in reply „We have included more details in the text on the few therapies currently available. We have included a graphical abstract/figure that summarizes this review”.
Unfortunately, I don't see the figure attached to the paper. Is it possible to show it?
Maybe it is contained in another file or sent separately? Either way, I think the work has been significantly revised, supplemented with more original work, and in this version it can be published.
If there were any minor proofreading errors (extra spaces, missing periods, etc.) that I did not notice, I hope they will "disappear" after the final brush-up.
We thank the reviewer for their careful reading of our revised manuscript. We have carefully proofread and edited the errors. We apologize for the graphical abstract exclusion in the revised template; we have re-added it to this revised version (2.0). It is now included before the Introduction.